**Subject Category:**
Biology (whole organism)

evolution/palaeontology

Insecta, Palaeodictyoptera, Megasecoptera, nymph, ecomorphology, tracheal respiratory system

**Author for correspondence:**
Jakub Prokop
e-mail: jprokop@natur.cuni.cz

# Ecomorphological diversification of the Late Palaeozoic Palaeodictyopterida reveals different larval strategies and amphibious lifestyle in adults

Jakub Prokop[1], Ewa Krzemińska[2], Wiesław Krzemiński[2], Kateřina Rosová[1], Martina Pecharová[1], André Nel[3] and Michael S. Engel[4,5]

[1]Department of Zoology, Faculty of Science, Charles University, Viničná 7, CZ-128 00, Praha 2, Czech Republic
[2]Institute of Systematics and Evolution of Animals, Polish Academy of Sciences, ul. Sławkowska 17, 31-016 Kraków, Poland
[3]Institut de Systématique, Évolution, Biodiversité, ISYEB - UMR 7205 – CNRS, MNHN, UPMC, EPHE, Muséum national d'Histoire naturelle, Sorbonne Universités, 57 rue Cuvier, CP 50, Entomologie 75005, Paris, France
[4]Division of Entomology, Natural History Museum, and Department of Ecology and Evolutionary Biology, University of Kansas, 1501 Crestline Drive – Suite 140, Lawrence, KS 66045, USA
[5]Division of Invertebrate Zoology, American Museum of Natural History, Central Park West at 79th Street, New York, NY 10024-5192, USA

JP, 0000-0001-6996-7832

The Late Palaeozoic insect superorder Palaeodictyopterida exhibits a remarkable disparity of larval ecomorphotypes, enabling these animals to occupy diverse ecological niches. The widely accepted hypothesis presumed that their immature stages only occupied terrestrial habitats, although authors more than a century ago hypothesized they had specializations for amphibious or even aquatic life histories. Here, we show that different species had a disparity of semiaquatic or aquatic specializations in larvae and even the supposed retention of abdominal tracheal gills by some adults. While a majority of mature larvae in Palaeodictyoptera lack unambiguous lateral tracheal gills, some recently discovered early instars had terminal appendages with prominent lateral lamellae like in living damselflies, allowing support in locomotion along with respiratory function. These results demonstrate that some species of Palaeodictyopterida had aquatic or semiaquatic

larvae during at least a brief period of their post-embryonic development. The retention of functional gills or gill sockets by adults indicates their amphibious lifestyle and habitats tightly connected with a water environment as is analogously known for some modern Ephemeroptera or Plecoptera. Our study refutes an entirely terrestrial lifestyle for all representatives of the early diverging pterygote group of Palaeodictyopterida, a greatly varied and diverse lineage which probably encompassed many different biologies and life histories.

## 1. Introduction

While the fossil record of hexapods extends to the Early Devonian, the first tangible evidence of aquatic insect specializations is documented from the Early Permian in some stem mayflies of Permoplectoptera and stoneflies (Plecoptera) [1–3]. Although the prior records of stem-group representatives of Ephemeroptera and Odonata with putatively aquatic larvae, are documented since the Late Carboniferous [4], the evidence is uncertain, perhaps with the exception of a meganeurid griffenfly larva *Dragonympha srokai* bearing lateral abdominal tracheal gills from the Late Carboniferous of the Mazon Creek 'Konservat-Lagerstätte' [5]. Based on their morphological specializations, most aquatic Permian immature and adult insects indicate lotic palaeoenvironmental conditions while the evidence from lentic habitats is lacking prior to the late Permian [6,7]. Wootton [6] provided extensive review on historical ecology of aquatic insects and considered the first Permian aquatic insects as predaceous. Kukalová [8] suggested a link between prognathous head and well-developed mandibles in early Permian larvae of Protereismatidae considering their predatory behaviour as well.

The rise of aquatic freshwater ecosystems since the Middle–Late Triassic is documented in several insect orders (e.g. Ephemeroptera, Plecoptera, Heteroptera, Coleoptera, Diptera) showing various ecomorphological specializations including those for stagnant lake habitats [6]. Wootton [6] considered that the increased role of insects in lake ecosystems, such as aquatic Heteroptera and Coleoptera, was due to the appearance of aquatic macrophytes in the early Mesozoic.

During the Late Palaeozoic, some of the most diverse, and morphologically and potentially biologically varied insects were among the superorder Palaeodictyopterida, a lineage including the earliest Pterygota ever found [9,10] and comprising four extinct orders: Palaeodictyoptera, Megasecoptera, Diaphanopterodea and Dicliptera [3,11]. While immature stages of the latter two orders are unknown, larvae are known from a variety of families of the former two orders and of varied morphotypes (figures 1 and 2). Not surprisingly, an understanding of their life history remains muddled and has been debated for over a century. Early authors such as Brongniart [12] considered immatures and even adults of some species to be amphibious, such as the megasecopteran *Corydaloides scudderi*, based on the presence of nine pairs of lateral abdominal bifid structures interpreted as tracheal gills. This view was followed by Brauer [13] and in some respect also by Handlirsch [14], who interpreted them either as abdominal tracheal gills or homologous structures. On the other hand, Lameere [15,16] doubted this interpretation and instead compared these prominent bifid structures with the lateral edges of terga of the extant mayfly *Oniscigaster wakefieldi*. This interpretation of such structures representing tergal projections was accepted by Carpenter [17] and followed by Kukalová-Peck [18], among others. It has been a dominant hypothesis in recent decades. However, these latter authors reported at the same time in megasecopterans, unusual 'long integumental projections covered with setae', 'dense dorsal outgrowths', or even 'very long filamentous structures' on the abdomen (e.g. Carpenter & Richardson [19, fig. 12, 13], Kukalová-Peck [20, fig. 31], Shear & Kukalová-Peck [21, fig. 18]). Wootton [22, p. 672] discussed with caution the life history of palaeodictyopteran larvae described from the UK and stated, '… not demonstrably terrestrial, but show no aquatic adaptations'. A weak position adopted by most past and even present researchers has been the assumption that the presumed biology of one or a few representative species was equally applicable to all of those within an extinct family or even across an entire order. While such an assumption may at times be justified, and certainly examples do exist among insects, there are even more cases in which the biologies of related genera and families differ greatly (e.g. Grimaldi & Engel [3]). Thus, the considerable variety of morphological specializations among Palaeodictyopterida probably reflects considerable biological diversity. The resolution of particular biologies among a variety of Palaeodictyopterida is needed in order to more fully understand the ecological breadth and success of insects during their first major diversification and prior to the cataclysmic extinction at the end of the Palaeozoic [23] and whether it was analogous to the tremendous success in virtually every terrestrial and freshwater habitat present today.

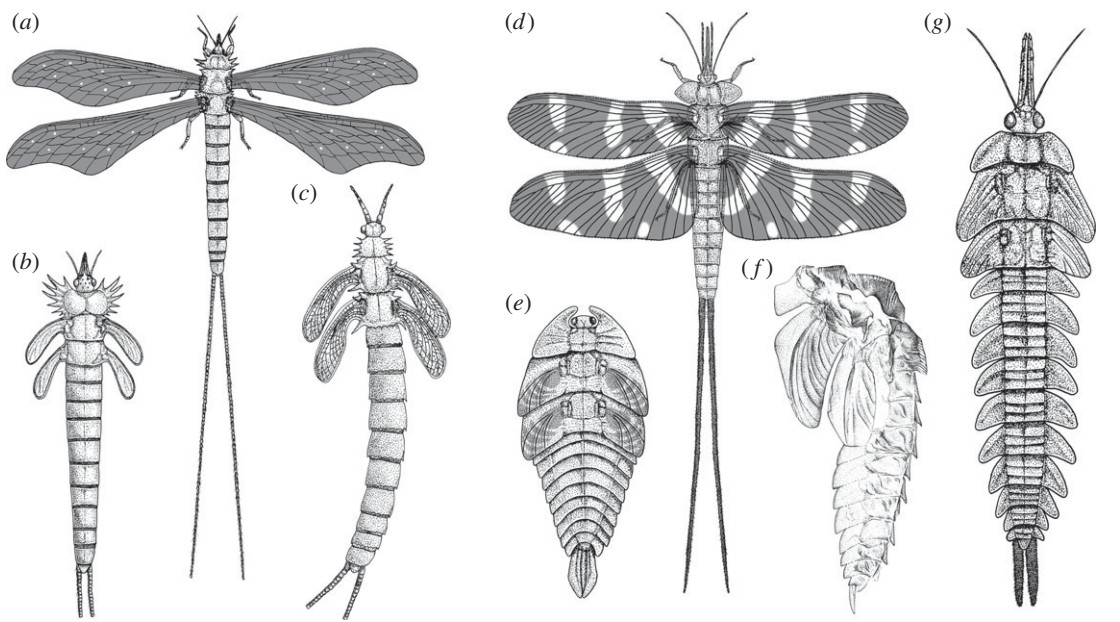

**Figure 1.** Disparity of morphotypes among larvae and adults of Late Carboniferous and Early Permian Megasecoptera and Palaeodictyoptera. (*a*) *Mischoptera nigra*, adult, Commentry, France, MNHN R51060. (*b,c*) *Mischoptera douglassi*. (*b*) Early larva of megasecopteran, reconstruction of habitus, FM PE31976, Mazon Creek, USA. (*c*) Reconstruction of older instar larva habitus, Douglass coll., Mazon Creek, USA. (*d*) *Dunbaria fasciipennis*, adult palaeodictyopteran, reconstruction of male habitus based on several specimens, Elmo, USA. (*e*) *Idoptilus* sp., Palaeodictyoptera, larval exuvia of early instar, no. GLAHM A.2680a, Stainborough, Barnsley, South Yorkshire, UK. (*f*) Palaeodictyoptera family indet., larval exuvia, ISEZ PAN IF-MP-1488-29-08, Upper Silesian Coal Basin, Sosnowiec—Klimontów, Poland. (*g*) *Bizarrea obscura*, palaeodictyopteran larva, reconstruction of habitus, FM PE11269, Mazon Creek, USA. (*a–e,g*) drawn by MP, (*f*) drawn by ZČ.

This issue is made all the more interesting as the phylogenetic relationships of the main pterygote insect lineages remain controversial, often dubbed the 'Palaeoptera problem' [24]. Palaeodictyopterida are traditionally considered as an early diverging group of Pterygota, either as the sister group to a putatively monophyletic Palaeoptera (the group comprising extant dragonflies and mayflies and their fossil relatives) [5,25], or resolved as sister group to Neoptera (all flying insects with the ability to fold their wings over the abdomen) on the basis of a recent phylogenetic analysis [26]. Interestingly, Palaeodictyopterida have not been recovered as sister group to all other Pterygota, i.e. to a Palaeoptera + Neoptera clade. The re-examination of wing base structures in a palaeodictyopteran genus, *Dunbaria* Tillyard (Spilapteridae), uncovered a mosaic of characters and identifies homologous structures to Odonatoptera, Ephemeropterida and also Neoptera [27]. Moreover, recent study their larval wing pad joints also supported the dual model of insect wing origin [28]. Accordingly, the diversity, biology and evolutionary history of Palaeodictyopterida are critical to understand Late Palaeozoic insect success.

The aim of this article is to critically evaluate the available morphological evidence on immature stages and selected adults of Palaeodictyopterida, and explore the plausibility of different hypotheses about their lifestyle strategies.

# 2. Material and methods

## 2.1. Specimen imaging and reconstruction

The results were obtained by comparison of the selected morphological structures between fossil and recent taxa using stereo microscopy. The specimens were observed under Zeiss Discovery V20 and Nikon SMZ1500 stereomicroscopes in a dry state and rarely under a film layer of ethyl alcohol. Photographs were taken with a Canon D550 digital camera, with MP-E 65 mm and EF 50 mm lenses. The original photographs were processed using Adobe Photoshop CS6, and for some images the focus-stacking software Helicon Focus Pro and Zerene Stacker were used. A few samples preserved with high three-dimensional relief were additionally examined with a Keyence VHX VH-Z20UR digital microscope.

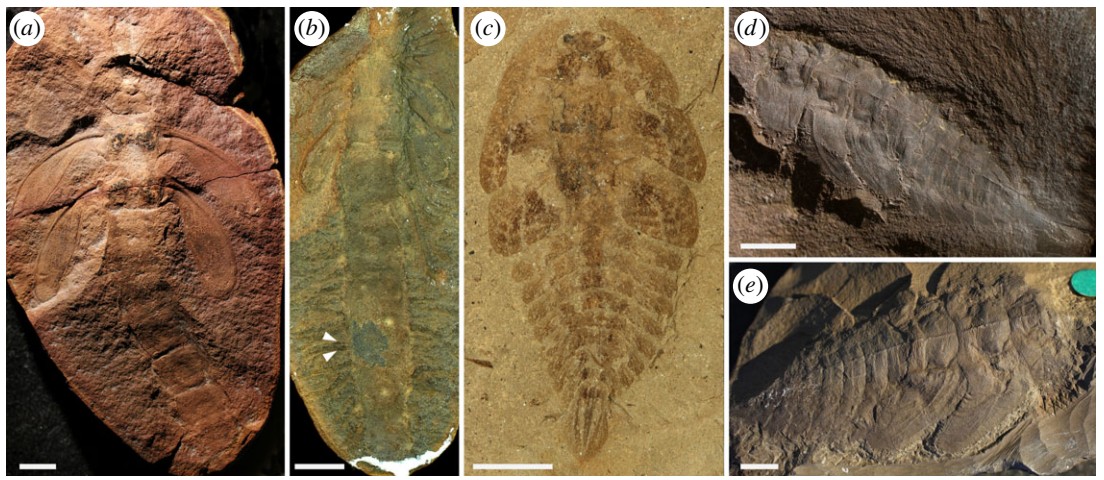

**Figure 2.** Larvae of Megasecoptera and Palaeodictyoptera. (*a*) *Lameereites* sp., Brodiidae, Megasecoptera, ROM no. 45546, Mazon Creek, IL, USA. (*b*) *Mischoptera douglassi*, Mischopteridae, Megasecoptera, HTP no. 1232, Mazon Creek, IL, USA. Arrows indicate filamentous projections as supposed tracheal gills. (*c*) *Idoptilus* sp., Palaeodictyoptera, larval exuvia of early instar, no. GLAHM A.2680a, Stainborough, Barnsley, South Yorkshire, UK. (*d*) *Rochdalia parkeri*, holotype MM L.11464, Lower Coal Measures, Rochdale, Lancashire, UK. (*e*) *Idoptilus onisciformis*, palaeodictyopteran larva, NHM In 44654, Middle Coal Measures of Barnsley, UK. Scale bars, (*a–e*) 5 mm. (*a*) Copyright © Royal Ontario Museum, Toronto, (*e*) copyright © Natural History Museum, London.

## 2.2. Material

Institutional abbreviations: FM, The Field Museum (Chicago, USA); GLAHM, Hunterian Museum, University of Glasgow (Glasgow, UK); HTP, Helen & Ted Piecko coll. (Chicago, USA); ISEZ PAN: Natural History Museum of the Institute of Systematics and Evolution of Animals PAS (Cracow, Poland); MM, Manchester Museum (Manchester, UK); MNHN, Muséum national d'Histoire naturelle (Paris, France); ROM, Royal Ontario Museum (Toronto, Canada); NHM, The Natural History Museum (London, UK); NMP, National Museum (Praha, Czech Republic); PIN, Paleontological Institute, Russian Academy of Sciences (Moscow, Russia); TS, Tomáš Soldán coll., Biology Centre, Czech Academy of Sciences, Institute of Entomology (České Budějovice, Czech Republic); YPM, Peabody Museum of Natural History, Yale University (New Haven, USA).

Abbreviations used for morphological structures are: bc, banded pattern of coloration, with alternating light and dark stripes; ds, dorsal spines; ma, precursor of convex vein MA; oe, broad outer edge of sheath (a developing wing); ps, pronotal spines; pw, prothoracic winglets; sp, spiracles; tg, tracheal gills.

## 3. Results and discussion

In this study, we thoroughly revise available and new fossils of megasecopteran and palaeodictyopteran larvae, as well as some adults to clarify morphological specializations and to see what evidence can be brought to support terrestrial, amphibious or even aquatic life histories. The disparity of larval habitus was considerable among these lineages, from elongate slender body forms with a spined prothorax in some megasecopterans to robust onisciform larvae (broad and flattened) often bearing prominent prothoracic winglets in palaeodictyopterans (figures 1 and 2). Developing wings in Palaeodictyopterida are well distinguished by the characteristic pattern of tracheal pleating and lacunal channels, including the presence of a clearly convex precursor of vein MA (figure 3*b,f*).

Wootton [22] studied palaeodictyopteran larvae from Carboniferous ironstone nodules found in the UK of the species *Idoptilus onisciformis* and *Rochdalia parkeri*, both sharing an onisciform body, the presence of prothoracic lobes or winglets and prominent abdominal laterotergites. However, he could not demonstrate any aquatic specializations present on these immatures. Re-examination of another larva from Middle Coal Measures in South Yorkshire, UK, assigned to *Idoptilus* sp. revealed early instar exuvia bearing articulated wing pads with three markedly convex ridges corresponding to the precursors of veins of RA, MA and CuA [29]. However, this young larva differed from all previously described Palaeodictyoptera by the presence of three triangular caudal appendages with prominent lateral lamellae densely covered by fine cuticular setae (figures 1*e* and 2*c*). It was assumed that these lamellae functioned

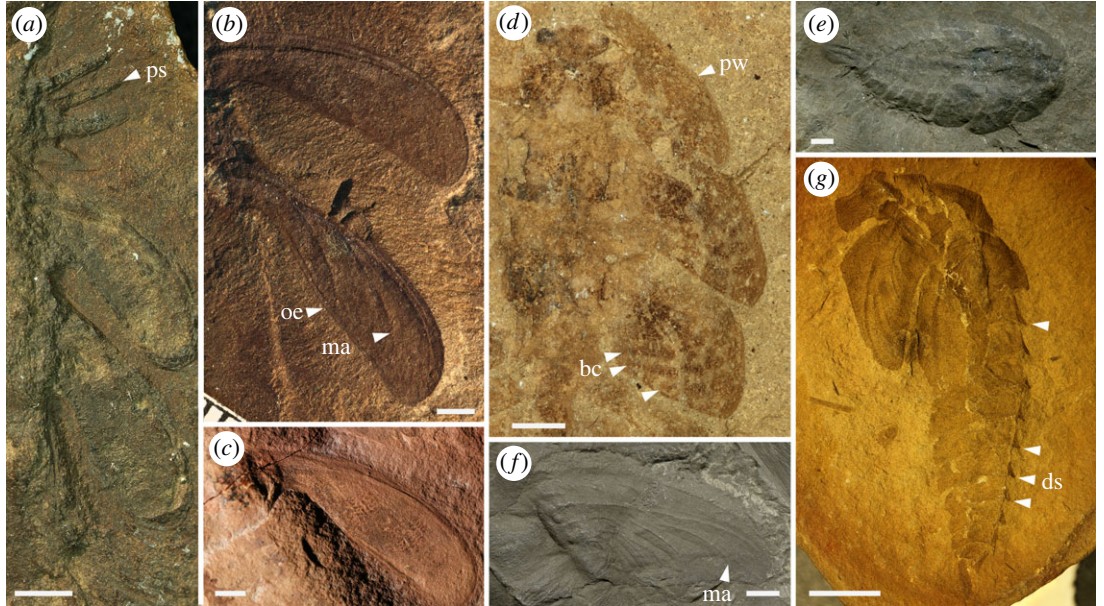

**Figure 3.** Ecomorphological specializations of megasecopteran and palaeodictyopteran larvae. (*a*) *Mischoptera douglassi*, Mischopteridae, Megasecoptera, HTP 1232, Late Carboniferous, Mazon Creek, IL, USA, detail of prothoracic lateral spines, meso- and metathoracic wing pads. (*b*) *Lameereites* sp., Brodiidae, Megasecoptera, YPM 66, Mazon Creek, IL, USA, detail of meso- and metathoracic wing pads. (*c*) *Lameereites* sp., Brodiidae, Megasecoptera, ROM 45546, Moscovian, Mazon Creek, IL, USA, detail of metathoracic wing pad. (*d,e*) *Idoptilus* sp., Palaeodictyoptera, GLAHM A.2680a, Stainborough, Barnsley, South Yorkshire, UK. (*d*) Wing pads, arrows indicate contrasting light- and dark-striped pads which allowed crypsis. (*e*) Habitus of larva in dorsolateral view. (*f*) *Idoptilus onisciformis*, holotype NHM In 44654, Middle Coal Measures of Barnsley, UK, detail of metathoracic wing pad. (*g*) Palaeodictyoptera family indet., larval exuvia, ISEZ PAN I-F-MP-1488-29-08, Upper Silesian Coal Basin, Sosnowiec—Klimontów, Poland. bc, banded pattern of coloration, with alternating light and dark stripes; ds, dorsal spines; ma, precursor of vein MA; oe, broad outer edge of sheath (a developing wing); ps, pronotal spines; pw, prothoracic winglet. Scale bars, (*a–f*) 2 mm; (*g*) 5 mm. (*c*) Copyright © Royal Ontario Museum, Toronto, (*e*) copyright © Natural History Museum, London.

as tracheal caudal gills owing to their structural resemblance to the gills of damselfly larvae (Odonata: Zygoptera), which are also used in the locomotion as, for instance, rudders and to escape from predators [30,31]. Another important difference found in the early larva of *Idoptilus* is the relatively shorter lamellae to body size, versus the comparatively larger lamellae known in the majority of extant damselfly larvae. The smaller surface area of gill lamellae corresponds with limited oxygen uptake; however, this could have been compensated by higher levels of atmospheric oxygen during the Late Palaeozoic [32], hence more dissolved oxygen in the water. Along with an ability to breathe underwater via caudal tracheal gills, the hydrodynamic constraints of larval body shape should be considered. The onisciform larvae like those of *Idoptilus* and *Rochdalia* show enlarged prothoracic lobes protecting the head, and broad abdominal laterotergites with pointed apices resembling aquatic larvae of water-penny beetles (Psephenidae) inhabiting fast-flowing streams [33] or some marsh beetles (Scirtidae) and Torridincolidae [34]. However, a similar habitus is also found in semiterrestrial or terrestrial insects, e.g. moss bugs (Coleorrhyncha: Peloridiidae), larvae of carrion beetles (Silphidae) and trilobite beetles (Lycidae: *Duliticola* sp.) (e.g. [35]). By contrast, the body shape of megasecopteran larvae attributed to *Mischoptera* sp. resemble Permian Protereismatidae (stem Ephemeroptera) with oblique, laterally positioned wing pads and a slender abdomen bearing prominent cerci but lacking a terminal filament. While the presence of tracheal gills was not unequivocally confirmed in this group, the supposed long projections with unknown function were reported and illustrated from early larval instars ([20, fig. 31], figure 2*b*).

Another remarkable feature on the wing pads of early instar of *Idoptilus* sp. were the prominent transversely banded pattern of coloration, with alternating light and dark stripes (figure 3*d*). Such a pattern of circular contrasting light and dark stripes is known in early larval instars of some extant damselflies and dragonflies (e.g. Calopterygidae, Aeshnidae, Epiophlebiidae) which live near the water surface among vegetation [36], although such patterns may also indicate occurrence in any environment, aquatic or terrestrial, with mottled light passing through overhead foliage. Wesenberg-Lund [37] considered such a pattern as protective and a kind of crypsis among mottled light resulting in the animal blending into the background.

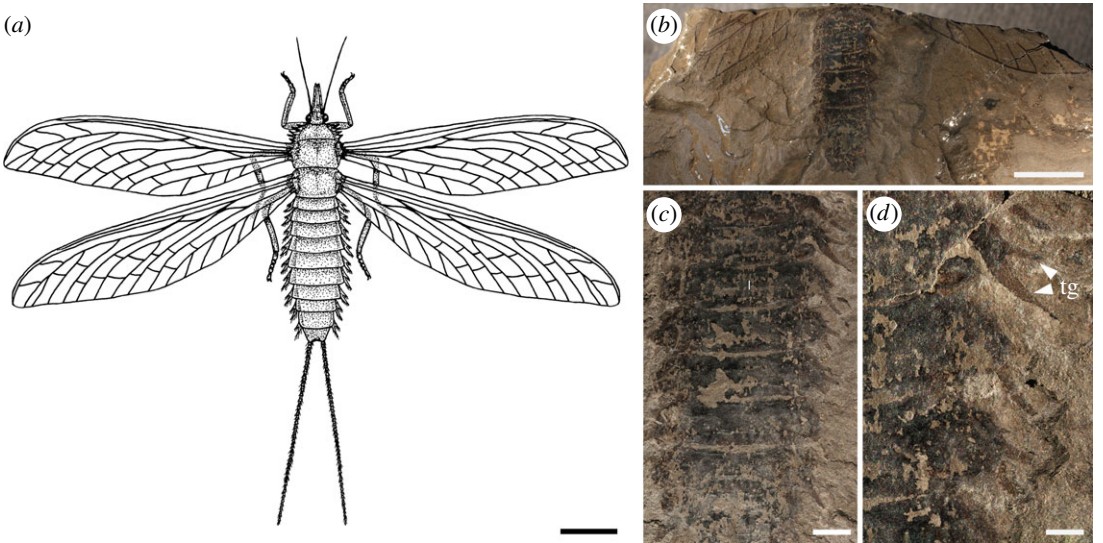

**Figure 4.** *Corydaloides scudderi* (Megasecoptera: Corydaloididae), Late Carboniferous, Commentry, France. (*a–d*) Image shows nine pairs of bifid abdominal structures with anterior and posterior parts pointed apically (presumably tracheal gills). (*a*) Reconstruction of habitus based on several specimens MNHN R51251, R51252, R51257, R51231 (drawn by MP). (*b*) Photograph of adult MNHN R51251. (*c,d*) Detail photographs of abdomen showing bifid tracheal gills and outlines of laterotergites MNHN R51251. tg, tracheal gills. Scale bars (*a,b*) 10 mm; (*c*) 2 mm; (*d*) 1 mm.

While these latter traits partly suggest an aquatic or semiaquatic lifestyle for early instars in some lineages of Palaeodictyoptera, particularly due to their lamellate form of caudal gills, other authors like Norling [38] placed less importance for respiration on caudal structures relative to lateral abdominal gills. Moreover, studies on extant dragonflies and damselflies have demonstrated that the caudal appendages can change greatly in form and presumably function during ontogeny [39], and this could have been the same in Palaeodictyoptera and could explain why older larvae completely lack such appendages.

An examination of later-stage palaeodictyopteran larvae from the Late Carboniferous of Sosnowiec, Poland shows the presence of prominent dorsal and lateral spines on abdominal segments, which most likely functioned for defence (figures 1*f* and 3*g*), as is found in many larvae of extant dragonflies and mayflies [40]. Anqvist & Johansson [41] demonstrated phenotypic adaptation of growth defensive spine trajectories in *Leucorhinia dubia* (Libellulidae) resulting from ontogenetic acceleration in environments with fish and leading to exaggerated spine shapes. If a similar predator influence and developmental mode was present in some lineages of Palaeodictyoptera, then this could be further indirect evidence for an aquatic lifestyle for the larvae of these genera.

The characteristic haustellate mouthparts of adult Palaeodictyopterida for sucking liquids is scarcely evident and rarely preserved in larvae of Palaeodictyoptera and Megasecoptera, but our limited available evidence supports a comparable form and function between larvae and adults as known for instance in *Mischoptera* sp. [42,43] (figure 1*a,b*). This has been one line of evidence used to suggest that larvae inhabited terrestrial habitats like their adults, and perhaps feeding on the same host plants. However, an amphibious lifestyle for larvae and adults cannot be entirely ruled out and there are analogues of such cases known in some modern stoneflies [44]. In this context, it is noteworthy that the Late Carboniferous ecosystems are reconstructed with high humidity and climate with indistinct seasonality. Interestingly, modern stonefly larvae and adults of Diamphipnoidae, such as *Diamphipnopsis samali*, preferably inhabiting habitats like riparian forests or waterfalls often retain abdominal tracheal gills as adults [45]. Admittedly, Palaeodictyopterida are distantly related to stoneflies and there is, therefore, no reason to assume the biology of the two would be identical. Nonetheless, ecological analogues to such a life history are known among some insect groups which independently evolved aquatic and semi-aquatic habits.

Štys & Soldán [46] demonstrated the retention of abdominal and accessory tracheal gills in subimagoes and adults of extant Ephemeroptera and reviewed the evidence in adults of Odonata, Plecoptera and other lineages with aquatic immatures. The re-examination of the megasecopteran adult *Corydaloides scudderi* (MNHN R51251) shows prominent lateral bifid structures as presumably nine pairs of abdominal tracheal gills with anterior and posterior parts pointed apically and emerging distinctly separated from the terga (figure 4*a–d*). Thus, at least *C. scudderi* retained presumably

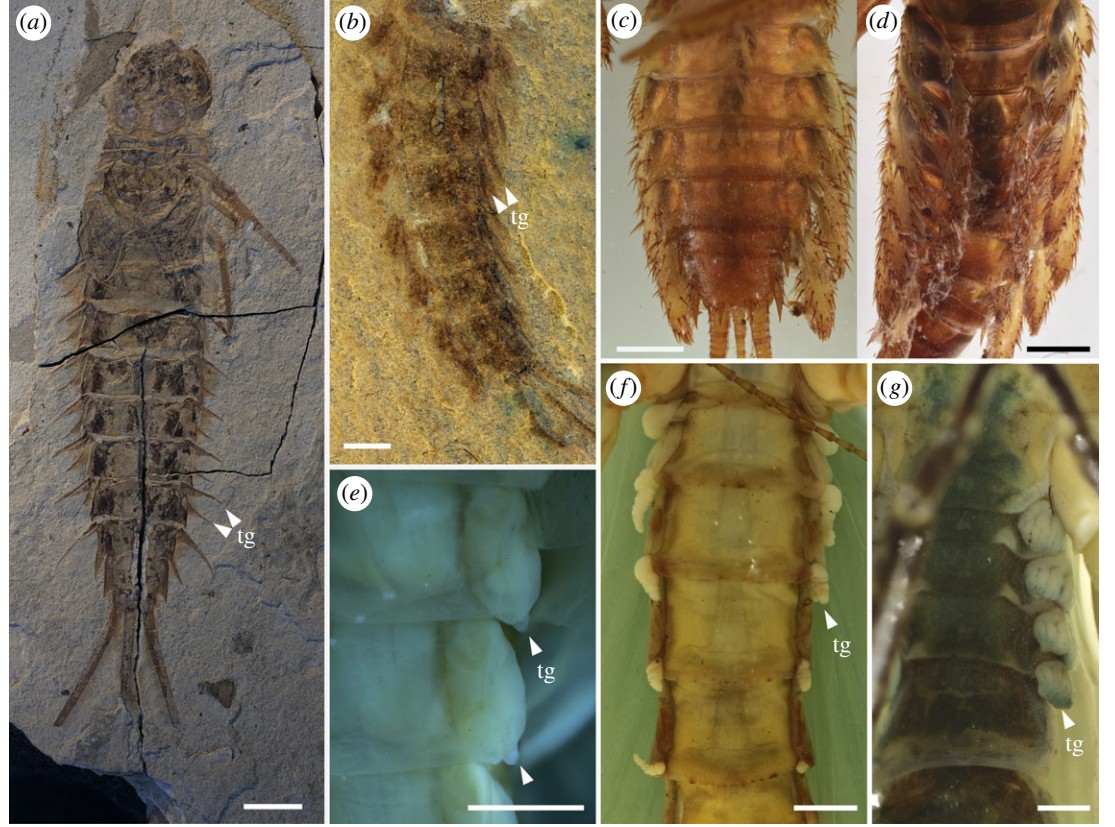

**Figure 5.** Abdominal tracheal gills in fossil and recent larvae of Ephemeroptera and retention of gills by their subimagoes and imagoes of Plecoptera. (*a–d*) Photographs of abdomen showing bifid tracheal gills and outlines of laterotergites. (*a*) *Ephemeropsis trisetalis*, Hexagenitidae, Ephemeroptera, PIN 3064-3332, Early Cretaceous, Baissa, Russia, larva with seven pairs of bifid abdominal tracheal gills. (*b*) *Misthodotes sharovi*, Mistodotidae, PIN 1700-374, Early Permian, Tshekarda, Russia, abdomen of nymph with discernable abdominal tracheal gills. (*c,d*) *Coloburiscus humeralis*, Coloburiscidae, Ephemeroptera, larva, TS coll., Cartenbury, New Zealand. (*e*) *Palingenia longicaudata*, Palingenidae, subimago, TS coll., Hungary. (*f*) *Neuroperla schedingi*, Eustheniidae, Plecoptera, NMP coll., IX. La Araucanía Region, Chile, imago, ventral aspect of abdomen with discernable tracheal gills. (*g*) *Diamphipnoa annulata*, Diamphipnoidae, Plecoptera, NMP coll., IX. La Araucanía Region, Chile, imago, ventral aspect of abdomen with discernable tracheal gills. tg, tracheal gills. Scale bars (*a*) 5 mm; (*b–g*) 1 mm.

abdominal tracheal gills as adults. However, we cannot definitely exclude other functions for these prominently bifid structures, as for example defensive or thermoregulatory. On the other hand, we assume that such structures are more commonly rigidly fixed and derived from the tergum or sternum, unlike what we observe in *C. scudderi*. These structures were previously highlighted by Brongniart [12] and compared with the gills in the modern stonefly *Pteronarcys regalis*, but overlooked in most recent studies.

The significance of these structures resides in their general structural resemblance to the gills of mayfly larvae, although in crown-group Ephemeroptera there is a maximum of seven pairs known versus the occurrence on abdominal segments I–IX (figure 5*a–d*). Interestingly, similar nine pairs of gills occurred in some stem-group Protereismatida, coeval with Permian Palaeodictyopterida (figure 5*b*) [4,47]. The structurally bifid gills in *C. scudderi* probably had spinose lamellae owing to the presence of numerous tubercles on their surface (figure 4*c,d*). Various types of setose structures on the ventral part of the gill lamellae have been experimentally demonstrated for a modern mayfly larva, *Epeorus assimilis* (Heptageniidae), to contribute to the friction coefficient and be used as underwater attachments in the current [48]. A form of gills in *C. scudderi* can be found for instance in modern mayfly larvae of the family Coloburiscidae (figure 5*c,d*). These spinose structures on the gills serve to anchor the larva beneath rocks in stony upland streams and for protection [49,50]. Nevertheless, another important comparison of abdominal gills in *C. scudderi* should be focused on the imaginal functional gills found in Plecoptera especially among extant species of *Diamphipnoa* and *Neoperla* (figure 5*f,g*). The function of adult gills has been experimentally studied in the modern stonefly

*Diamphipnopsis samali* occurring in mountain stream habitats in Chile and Argentina [44]. In this species, the alternate contact of gills to air and water during rowing, and the ability of gills to contribute in gas exchange was demonstrated. While the structure of adult gills in extant stoneflies differs from *C. scudderi* and these lineages are by no means related, it may serve as an ecologically convergent analogue for interpreting a possible function in this fossil species. More importantly, some extant mayflies retain as subimagoes more similar bifid gills (figure 5*e*), such as *Palingenia longicaudata* (Štys & Soldán [46, p. 412, fig. 5]), although the function and operation of such gills in *P. longicaudata* have not yet been explored. If the lateral abdominal bifid structures represent functional tracheal gills in adults of *C. scudderi* we must consider the possibility that these insects obtained oxygen in shallow water edges, waterfalls, or perhaps when floating on the water surface and, owing to their wings, were probably never submerged. Indeed, the permanently outstretched wings in Megasecoptera would have been a major hindrance to a fully aquatic lifestyle in open and deep pools.

In general, it is difficult to infer the original habitat for these insects based on taphonomy, as the fossil record is poor with few isolated larvae, detached wing pads and larval exuviae (e.g. [22,51]). It is also necessary to keep in mind that the fossilized nymphs often occur in sedimentary rocks corresponding to palaeoenvironments in which they died and thus perhaps do not represent locations preferred in life. In addition, the exuviae are extremely prone to secondary transportation by wind and water currents due to their lightness. In the majority of localities, these larvae are found in the same assemblage together with adults. It is also the case for the Early Permian stem mayflies Protereismatidae and Misthodotidae where larvae, their exuviae, and isolated wing pads have been found in the same layers together with adults [8,52]. However, only from a few Pennsylvanian localities like Sosnowiec near Katowice (Silesia, Poland) and a single area in the Piesberg Quarry near Osnabrück (Lower Saxony, Germany) could we detect in taphocoenoses the marked abundance of palaeodictyopteran larval exuviae and detached wing-pad sheaths supporting the idea of their life took place inside or in close proximity to these aquatic/riparian habitats [53,54].

## 4. Conclusion

The biology of Palaeodictyopterida appears to have been more varied than has been long surmised, much like most orders of insects today. The evaluation of available data from the morphology of some larval stages as well as adults from a few species among Palaeodictyoptera and Megasecoptera reveals direct and indirect evidence for amphibious or possibly aquatic lifestyles in certain taxa. This can be determined from different aspects of external morphology as mainly the presence of caudal tracheal gills in early larval instars and most importantly retention of rudimentary or functional tracheal abdominal gills by adults. We, therefore, presume that at least these genera were amphibious or aquatic in early larval stages, transitioning possibly into a semiaquatic mode in mature larvae (much like petalurid dragonfly larvae (e.g. *Petalura gigantea*)) that live in flooded burrows during the day and forage on land during the night tolerating aerial conditions [55,56], and possibly even an amphibious lifestyle in some adults. Naturally, it cannot be assumed that such biology was fixed across these extinct orders, as the extreme morphological variety (including the absence of gill-like structures) demonstrates that these lineages had diversified into a considerable number of niches, analogous to modern orders such as Heteroptera in which both terrestrial and aquatic lineages coexist. It remains to be discovered what habits were most prevalent (the larvae and biology of most species of Palaeodictyopterida remain unknown), or what mode of life pre-dated the Carboniferous appearance of these lineages. Both terrestrial and aquatic genera may be found within individual families of insects, and it should therefore perhaps not be surprising that the Palaeozoic insect fauna exhibited a similar ecological breadth, one which took advantage of diverse habitats in both the water and on land.

Data accessibility. Specimens are accessible in the following collections: The Field Museum (Chicago, USA), Hunterian Museum, University of Glasgow (Glasgow, UK); Helen & Ted Piecko coll. (Chicago, USA), Natural History Museum of the Institute of Systematics and Evolution of Animals PAS (Cracow, Poland), Manchester Museum (Manchester, UK), Muséum national d'Histoire naturelle (Paris, France); Royal Ontario Museum (Toronto, Canada), The Natural History Museum (London, UK), National Museum (Prague, Czech Republic), Paleontological Institute, Russian Academy of Sciences (Moscow, Russia), and Peabody Museum of Natural History, Yale University (New Haven, USA). All other data are listed in figures.
Authors' contributions. J.P. conceived the initial idea following valuable debate together with E.K. and W.K. and designed the project. J.P., A.N. and M.S.E. drafted the manuscript, to which all authors contributed. J.P., K.R., M.P., A.N., E.K., W.K. and M.S.E. participated in the morphological studies, analysed the data and commented on the manuscript. All authors approved the final draft.
Competing interests. We declare we have no competing interests.

Funding. J.P. and M.P. were supported by a research project of the Grant Agency of the Czech Republic (no. 18-03118S), while K.R. was supported by Charles University Grant Agency no. 1612218 PřF B-BIO.

Acknowledgements. We are indebted to the scientific illustrator Zuzana Čadová (ZC) for the magnificent line drawing of palaeodictyopteran larval exuvia found in Sosnowiec (Poland). The authors are grateful to Susan Butts and Jessica Utrup (both Peabody Museum of Natural History, Yale University, New Haven, USA), Neil Clark (Hunterian Museum, Glasgow, UK), Angelika Leipner (Museum am Schölerberg, Osnabrück, Germany), Claire Mellish (The Natural History Museum, London, UK), Nina Sinisthenkova and Alexander Rasnitsyn (both Paleontological Institute, Russian Academy of Sciences, Moscow, Russia), Peter Fenton and David Rudkin (both Royal Ontario Museum, Toronto, Canada), David Gelsthorpe (Manchester Museum, Manchester, UK), and Paul Mayer (The Field Museum, Chicago, USA), who kindly provided access to the collections under their care. J.P. is grateful to Martin Fikáček and Pavel Chvojka (both National Museum, Praha, Czech Republic) and Pavel Sroka (Institute of Entomology, Biology Centre, Czech Academy of Sciences, České Budějovice, Czech Republic) for the loan of mayflies and stoneflies allowing comparative study of abdominal tracheal gills, and to Lenka Váchová (National Museum, Praha, Czech Republic) for her help with Keyence digital microscopy. We are grateful to anonymous reviewers for their insightful comments on the early versions of the manuscript. We cordially thank Darek Wojciechowski for his efforts in fieldwork at Sosnowiec.

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
