## [Reviewer comments · Royal Society Open Science]

Review History

RSOS-190460.R0 (Original submission)

Review form: Reviewer 1

Is the manuscript scientifically sound in its present form?

Yes

Are the interpretations and conclusions justified by the results?

Yes

Is the language acceptable?

Yes

Is it clear how to access all supporting data?

Yes

Do you have any ethical concerns with this paper?

No

Have you any concerns about statistical analyses in this paper?

No

Recommendation?

Accept with minor revision (please list in comments)

Comments to the Author(s)

This is an interesting manuscript, sometimes the conclusions are very bold and some are far-fetched. I would suggest the authors read through it and may try to qualify their most important statements.

Normally, I'm not a big fan of stating the English needs revision because I'm not a native speaker myself. However, I feel like there are some parts with better and some with worse English. It feels like the last revision was made by a non-native speaker and some of the changes stand out. Since there is a native speaker under the authors I would suggest him to give the manuscript a thorough read.

However, generally I suggest publication after minor revision. Further comments are in the following:

Abstract

P7 L19 and following - I would hardly recommend avoiding terms like superorder or other ranks, we live in a time after the brilliant work of Henning so please act like it.

P7 L24 - "number of mature nymphs" is very unspecific

P8 L42 - "stem groups" I would prefer stem group representatives. However, the sentence is monstrous can you may divide it?

P8L47-48 - the transition from habitat to predation mode is rude ("... prior to the late Permian [6,7]. Wootton [6] considered ...") please add a nice connecting sentence.

P9L75 - "... not demonstrably terrestrial, but lacking aquatic adaptations" if this is a direct quote you would need the page number I suppose.

P9L76 - "A weak position ..." your absolutely right, but that's your interpretation of data, please leave this for the discussion.

P9L81 - "morphologies" morphological adaptations?

P9L81 - "The resolution of ..." this is an enormous sentence and very hard to follow please explain and simplify.

P10L86 - "This matter is made all the more interesting ..." What?

P10L98 - And? What are your questions? Any hypothesis? What can we expect in your paper? There is a substantial part of the introduction missing! Maybe you can find parts of it in the very first part of the results section ...

Method section

P10L101 - "... comparative morphological study ..." that's not a method, you compared the morphology using stereo microscopy

P10L103 - "Observations were made using a Zeiss Discovery V20 and Nikon SMZ1500 in dry state or under a film layer of ethyl alcohol." That sounds like you are using your microscopes under a film layer or ethanol.

Results/Discussion

P12L140 - "rudders" well if you see Zygoptera swimming the main power for the protraction is generated by the caudal gills. Of course, they use them for steering but not at all exclusively. Furthermore, there are numerous studies showing that Zygoptera do quite well without their gills.

P12L143 - "Recent" recent

P12L147 - "The onisciform nymphs ... inhabiting fast-flowing streams." citation is needed.

P12L150 - "However, a similar habitus ... (Lampyridae:Duliticola sp.)." citation is needed.

P13L183 - "[38,39]: see figures 1a,b]" something is wrong with the brackets.

P14L208 - "These spinose structures on the gills serve to anchor the nymph beneath rocks in stony upland streams and for protection. citation is needed.

For the last part of the discussion the paper on Ephemeroptera larvae of Ditsche-Kuru et al. 2010, JEB (doi: 10.1242/jeb.037218) might be of interest.

Figure 1 - I'm missing a scale bar, please add one if possible.

Review form: Reviewer 2 (Enrique Peñalver)

Is the manuscript scientifically sound in its present form?

Yes

Are the interpretations and conclusions justified by the results?

Yes

Is the language acceptable?

Yes

Is it clear how to access all supporting data?

Yes

Do you have any ethical concerns with this paper?

No

Have you any concerns about statistical analyses in this paper?

No

Recommendation?

Accept with minor revision (please list in comments)

Comments to the Author(s)

I read the comments of the previous referees 1 and 2 and I think that the authors improved the manuscript attending them. I only want comment about one response of the authors to a topic noted by the both referees. Authors indicated that: "... a banded pattern of coloration is not indicative of only aquatic environment ... that contemporary early instar nymphs of Odonata bear exactly the same circular banded pattern on wing pads which serve for their protection against the predators. In later instars this pattern disappears because nymphs are larger and having fewer risk of attacks. This can be seen on early instar nymphs of Palaeodictyoptera, so we think it is important to demonstrate this evidence in our text." Similar explanation was provided to referee 2. However, please note that this is a strongly "adaptationist explanation", but it can be analyzed under the constructional morphology model (e.g., see Briggs, D.E.G., "Seilacher on the Science of Form and Function"), mainly considering that this character is present in Palaeodictyoptera and other related groups during early instars. Maybe the authors could include a sentence about additional potential explanations. For example, it is not plausible that this pattern was highly-energy-consuming to maintain it during ontogeny, and it is not plausible that the hypothetical decrease of predation in mature instars implicated a high energy savings that would result, from natural selection, not to maintain it.

The main comment I want to include is that in this manuscript there is not mention of taphonomical evidence supporting or not the main topic of this manuscript. It is know that in several burial environments there is an overrepresentation of aquatic taxa, and etc. At this respect, I think that one of the authors could include some comments about this topic considering the fossil record of Palaeodictyoptera in general, because he investigated about taphonomy in aquatic environments (André Nel).

Minor corrections:

Line 5: "Engel," changes to "Engel"

Line 45: Lagerstätte" (outcrop in German) is not informative. Do the authors want to indicate "Konzentrar-Lagerstätte" or "Konservat-Lagerstätte"? I think that the latter.

Line 75: "...." Changes to "..."

Lines 93-94: "...genus, Dunbaria Tillyard (Spilapteridae), uncovered..." [note the two commas]

Line 129: maybe figures (plural). Please, revise this detail along the manuscript because it is not clear the guideline applied

Lines 131-132: "...onisciform body form..." changes to "...onisciform body..."

Line 134: "...Yorkshire, UK, assigned..." [note the extra comma]

Line 144: "have" changes to "has"

Line 170: please, revise "...this could have been be..."

Line 204: "... a similar nine pairs...", please revise the use of singular or plural in this expression

Line 226: "reveal" changes to "reveals"

Line 245: "follwing" changes to "following"

Lines 256-260: drawings and reconstructions are always interpretative. I think that this section must to include this contribution. At this respect, is evident that MP contributed, but who is ZC (apparently not a co-author) as indicated at the end of figure caption 1? If is not a co-author, please include his name in Acknowledgements.

Line 382: "Bull." in italics

Figure captions: the genus and species names are not in italics!

Figure caption 1: "U.S.A.." changes to "U.S.A." (two times). Note that for the same, the authors used indistinctly U.S.A. and USA. In line 27, apparently authors must indicate if the specimen is mature (subfigure c)

Figure caption 3: line 29: (d,e) maybe corresponds to (d,f), please revise it. It is not clear the taxon of subfigure f

Figure caption 4: line 24 the indication of subfigure a must be in parenthesis. Line 25: M.P. is the same author that MP in figure caption 1. Please, use the same format. It is not indicated that (c,d) are photographs of the same specimen that in (b). I deduced it.

Figure caption 5: in the title, please include "fossil and Recent" even that circumstance is evident. Line 33: the indication of subfigure a must be with parenthesis. The explanation of the subfigure (e) lacks indication of the outcrop and country.... Note that the correct spelling is "La Araucanía" (two times in this figure caption)

Decision letter (RSOS-190460.R0)

19-Jun-2019

Dear Dr Prokop

On behalf of the Editors, I am pleased to inform you that your Manuscript RSOS-190460 entitled "Ecomorphological diversification of the Late Paleozoic Palaeodictyoptera reveals different nymphal strategies and amphibious lifestyle in adults" has been accepted for publication in Royal Society Open Science subject to minor revision in accordance with the referee suggestions. Please find the referees' comments at the end of this email.

The reviewers and handling editors have recommended publication, but also suggest some minor revisions to your manuscript. Therefore, I invite you to respond to the comments and revise your manuscript.

- Ethics statement

- Data accessibility

<http://datadryad.org/submit?journalID=RSOS&manu=RSOS-190460>

- **Competing interests**

- **Authors' contributions**

- **Acknowledgements**

- **Funding statement**

Because the schedule for publication is very tight, it is a condition of publication that you submit the revised version of your manuscript before 28-Jun-2019. Please note that the revision deadline will expire at 00.00am on this date. If you do not think you will be able to meet this date please let me know immediately.

on behalf of Dr Robert Sansom (Associate Editor) and Kevin Padian (Subject Editor)
 openscience@royalsociety.org

Reviewer comments to Author:

Reviewer: 1

Comments to the Author(s)

This is an interesting manuscript, sometimes the conclusions are very bold and some are far-fetched. I would suggest the authors read through it and may try to qualify their most important statements.

Normally, I'm not a big fan of stating the English needs revision because I'm not a native speaker myself. However, I feel like there are some parts with better and some with worse English. It feels like the last revision was made by a non-native speaker and some of the changes stand out. Since there is a native speaker under the authors I would suggest him to give the manuscript a thorough read.

However, generally I suggest publication after minor revision. Further comments are in the following:

Abstract

P7 L19 and following - I would hardly recommend avoiding terms like superorder or other ranks, we live in a time after the brilliant work of Henning so please act like it.

P7 L24 - "number of mature nymphs" is very unspecific

P8 L42 - "stem groups" I would prefer stem group representatives. However, the sentence is monstrous can you may divide it?

P8L47-48 - the transition from habitat to predation mode is rude ("... prior to the late Permian [6,7]. Wootton [6] considered ...") please add a nice connecting sentence.

P9L75 - "... not demonstrably terrestrial, but lacking aquatic adaptations" if this is a direct quote you would need the page number I suppose.

P9L76 - "A weak position ..." your absolutely right, but that's your interpretation of data, please leave this for the discussion.

P9L81 - "morphologies" morphological adaptations?

P9L81 - "The resolution of ..." this is an enormous sentence and very hard to follow please explain and simplify.

P10L86 - "This matter is made all the more interesting ..." What?

P10L98 - And? What are your questions? Any hypothesis? What can we expect in your paper? There is a substantial part of the introduction missing! Maybe you can find parts of it in the very first part of the results section ...

Method section

P10L101 - "... comparative morphological study ..." that's not a method, you compared the morphology using stereo microscopy

P10L103 - "Observations were made using a Zeiss Discovery V20 and Nikon SMZ1500 in dry state or under a film layer of ethyl alcohol." That sounds like you are using your microscopes under a film layer or ethanol.

Results/Discussion

P12L140 - "rudders" well if you see Zygoptera swimming the main power for the protraction is generated by the caudal gills. Of course, they use them for steering but not at all exclusively. Furthermore, there are numerous studies showing that Zygoptera do quite well without their gills.

P12L143 - "Recent" recent

P12L147 - "The onisciform nymphs ... inhabiting fast-flowing streams." citation is needed.

P12L150 - "However, a similar habitus ... (Lampyridae:Duliticola sp.)." citation is needed.

P13L183 - "[38,39]: see figures 1a,b]" something is wrong with the brackets.

P14L208 - "These spinose structures on the gills serve to anchor the nymph beneath rocks in stony upland streams and for protection. citation is needed.

For the last part of the discussion the paper on Ephemeroptera larvae of Ditsche-Kuru et al. 2010, JEB (doi: 10.1242/jeb.037218) might be of interest.

Figure 1 - I'm missing a scale bar, please add one if possible.

Reviewer: 2

Comments to the Author(s)

I read the comments of the previous referees 1 and 2 and I think that the authors improved the manuscript attending them. I only want comment about one response of the authors to a topic noted by the both referees. Authors indicated that: "... a banded pattern of coloration is not indicative of only aquatic environment ... that contemporary early instar nymphs of Odonata bear exactly the same circular banded pattern on wing pads which serve for their protection against the predators. In later instars this pattern disappears because nymphs are larger and having fewer risk of attacks. This can be seen on early instar nymphs of Palaeodictyoptera, so we think it is important to demonstrate this evidence in our text." Similar explanation was provided to referee 2. However, please note that this is a strongly "adaptationist explanation", but it can be analyzed under the constructional morphology model (e.g., see Briggs, D.E.G., "Seilacher on the Science of Form and Function"), mainly considering that this character is present in Palaeodictyoptera and other related groups during early instars. Maybe the authors could include a sentence about additional potential explanations. For example, it is not plausible that this pattern was highly-energy-consuming to maintain it during ontogeny, and it is not plausible that the hypothetical decrease of predation in mature instars implicated a high energy savings that would result, from natural selection, not to maintain it.

The main comment I want to include is that in this manuscript there is not mention of taphonomical evidence supporting or not the main topic of this manuscript. It is know that in several burial environments there is an overrepresentation of aquatic taxa, and etc. At this respect, I think that one of the authors could include some comments about this topic considering the fossil record of Palaeodictyoptera in general, because he investigated about taphonomy in aquatic environments (André Nel).

Minor corrections:

Line 5: "Engel," changes to "Engel"

Line 45: Lagerstätte" (outcrop in German) is not informative. Do the authors want to indicate "Konzentrar-Lagerstätte" or "Konservat-Lagerstätte"? I think that the latter.

Line 75: "...." Changes to "..."

Lines 93-94: "...genus, Dunbaria Tillyard (Spilapteridae), uncovered..." [note the two commas]

Line 129: maybe figures (plural). Please, revise this detail along the manuscript because it is not clear the guideline applied

Lines 131-132: "...onisciform body form..." changes to "...onisciform body..."

Line 134: "...Yorkshire, UK, assigned..." [note the extra comma]

Line 144: "have" changes to "has"

Line 170: please, revise "...this could have been be..."

Line 204: "... a similar nine pairs...", please revise the use of singular or plural in this expression

Line 226: "reveal" changes to "reveals"

Line 245: "follwing" changes to "following"

Lines 256-260: drawings and reconstructions are always interpretative. I think that this section must to include this contribution. At this respect, is evident that MP contributed, but who is ZC (apparently not a co-author) as indicated at the end of figure caption 1? If is not a co-author, please include his name in Acknowledgements.

Line 382: "Bull." in italics

Figure captions: the genus and species names are not in italics!

Figure caption 1: "U.S.A." changes to "U.S.A." (two times). Note that for the same, the authors used indistinctly U.S.A. and USA. In line 27, apparently authors must indicate if the specimen is mature (subfigure c)

Figure caption 3: line 29: (d,e) maybe corresponds to (d,f), please revise it. It is not clear the taxon of subfigure f

Figure caption 4: line 24 the indication of subfigure a must be in parenthesis. Line 25: M.P. is the same author that MP in figure caption 1. Please, use the same format. It is not indicated that (c,d) are photographs of the same specimen that in (b). I deduced it.

Figure caption 5: in the title, please include "fossil and Recent" even that circumstance is evident.

Line 33: the indication of subfigure a must be with parenthesis. The explanation of the subfigure (e) lacks indication of the outcrop and country.... Note that the correct spelling is "La Araucanía" (two times in this figure caption)

Author's Response to Decision Letter for (RSOS-190460.R0)

See Appendix A.

RSOS-190460.R1 (Revision)

Review form: Reviewer 3 (Rolf Beutel)

Is the manuscript scientifically sound in its present form?

Yes

Are the interpretations and conclusions justified by the results?

Yes

Is the language acceptable?

Yes

Do you have any ethical concerns with this paper?

No

Have you any concerns about statistical analyses in this paper?

No

Recommendation?

Accept with minor revision (please list in comments)

Comments to the Author(s)

This is an exceptionally interesting topic. The findings and interpretations are of great relevance for the understanding of the early evolution of pterygote insects. The authors are obviously very competent.

l. 88: I would reword this to make it less ambiguous ("Palaeoptera problem").

. 97: "uncovered a mosaic of characters and numerous homologies to Odonatoptera, Ephemeropterida, and also Neoptera [27]."

What exactly means this, the identification of homologous structures? Or potential synapomorphies? Please clarify.

l. 146: "...microstructures as setae..."

Something seems to be missing in the sentence.

l. 151. "damsfly nymphs"

These immatures have specific larval structures like the labial mask and the terminal appendages, so they are by definition larvae and not nymphs.

l. 157. Also in some larvae of Scirtidae and Torridincolinae (e.g. Beutel et al. 1998 or Handbook of Zoology Coleoptera, Vol. 1)

l. 199. This statement is awkward. It is evident that this stonefly species is not closely related to the extinct taxa under consideration. So it is obvious that the stonefly only indicates that this scenario is possible on principle.

As a whole this a well written study with very interesting results. In the present version the interpretations are presented carefully and cautiously. I recommend publication after very minor revision.

I recommend a final linguistic check by Dr. Engel.

Review form: Reviewer 4 (Andrew Ross)

Is the manuscript scientifically sound in its present form?

No

Are the interpretations and conclusions justified by the results?

No

Is the language acceptable?

Yes

Do you have any ethical concerns with this paper?

No

Have you any concerns about statistical analyses in this paper?

No

Recommendation?

Accept with minor revision (please list in comments)

Comments to the Author(s)

This is an interesting and thought provoking article that deserves publication, however there is a problem that the authors and reviewers have not addressed. I'm perfectly happy that palaeodictyopteroid nymphs could have been aquatic however I'm not convinced that adults could also have been amphibious and survived in an aquatic/semiaquatic environment.

1) Page 9, line 207. Why are the lateral structures 'tracheal gills'? What's the evidence? Why could they not be another sort of structure, either for defence or thermoregulation (as hypothesized for prothoracic lobes)? Why would an adult palaeodictyopterid need tracheal gills? Surely their wings would have been a major hindrance to an aquatic lifestyle.

2) Page 10, line 225. Certainly it is feasible that the gills of *Diamphipnopsis* could allow it to breath while rowing on the water, thus is a survival strategy, however Palaeodictyoptera would not have been able to row. Palaeodictyoptera, like dragonflies, had two pairs of outstretched wings, which unlike stoneflies, they could not fold along their backs. A dragonfly, if it finds itself in the predicament of being on the water surface is likely to die. It can't swim and can't fly from the water surface because it's wings are trapped by the surface tension (see 'Drowning Dragonfly' on You Tube). It's either going to drown or be eaten by a fish. Its only hope is to drift to the water's edge and be able to grab onto something to drag itself out (I don't know if this has ever been observed or if a water-logged dragonfly would have the strength to do this). I can imagine that if an adult palaeodictyopterid found itself in a similar predicament it would suffer a similar fate. The only possible advantage I can see to it having lateral gills is that while floating on the water surface it may be able to get more oxygen to prevent it from drowning and thus give it more time to drift to the water's edge and to safety (if it had the strength to pull itself out and if it didn't get eaten by a fish or amphibian first- there were some big ones around at that time).

Decision letter (RSOS-190460.R1)

31-Jul-2019

Dear Dr Prokop:

On behalf of the Editors, I am pleased to inform you that your Manuscript RSOS-190460.R1 entitled "Ecomorphological diversification of the Late Paleozoic Palaeodictyoptera reveals different nymphal strategies and amphibious lifestyle in adults" has been accepted for publication in Royal Society Open Science subject to minor revision in accordance with the referee suggestions. Please find the referees' comments at the end of this email.

The reviewers and Subject Editor have recommended publication, but also suggest some minor revisions to your manuscript. Therefore, I invite you to respond to the comments and revise your manuscript.

- Ethics statement

- Data accessibility

If you wish to submit your supporting data or code to Dryad (<http://datadryad.org/>), or modify your current submission to dryad, please use the following link:
<http://datadryad.org/submit?journalID=RSOS&manu=RSOS-190460.R1>

- Competing interests

- Authors' contributions

- Acknowledgements

- Funding statement

Because the schedule for publication is very tight, it is a condition of publication that you submit the revised version of your manuscript before 09-Aug-2019. Please note that the revision deadline will expire at 00.00am on this date. If you do not think you will be able to meet this date please let me know immediately.

Supplementary files will be published alongside the paper on the journal website and posted on

the online figshare repository (<https://figshare.com>). The heading and legend provided for each supplementary file during the submission process will be used to create the figshare page, so please ensure these are accurate and informative so that your files can be found in searches. Files on figshare will be made available approximately one week before the accompanying article so that the supplementary material can be attributed a unique DOI.

on behalf of Kevin Padian (Subject Editor)
openscience@royalsociety.org

Reviewer comments to Author:
Reviewer: 3

Comments to the Author(s)

This is an exceptionally interesting topic. The findings and interpretations are of great relevance for the understanding of the early evolution of pterygote insects. The authors are obviously very competent.

l. 88: I would reword this to make it less ambiguous (“Palaeoptera problem”).

. 97: “uncovered a mosaic of characters and numerous homologies to Odonatoptera, Ephemeropterida, and also Neoptera [27].”

What exactly means this, the identification of homologous structures? Or potential synapomorphies? Please clarify.

l. 146: “...microstructures as setae...”

Something seems to be missing in the sentence.

l. 151. “damselfly nymphs”

These immatures have specific larval structures like the labial mask and the terminal appendages, so they are by definition larvae and not nymphs.

l. 157. Also in some larvae of Scirtidae and Torridincolinae (e.g. Beutel et al. 1998 or Handbook of Zoology Coleoptera, Vol. 1)

l. 199. This statement is awkward. It is evident that this stonefly species is not closely related to the extinct taxa under consideration. So it is obvious that the stonefly only indicates that this scenario is possible on principle.

As a whole this a well written study with very interesting results. In the present version the interpretations are presented carefully and cautiously. I recommend publication after very minor revision.

I recommend a final linguistic check by Dr. Engel.

Reviewer: 4

Comments to the Author(s)

This is an interesting and thought provoking article that deserves publication, however there is a problem that the authors and reviewers have not addressed. I'm perfectly happy that palaeodictyopteroid nymphs could have been aquatic however I'm not convinced that adults could also have been amphibious and survived in an aquatic/semiaquatic environment.

1) Page 9, line 207. Why are the lateral structures 'tracheal gills'? What's the evidence? Why could they not be another sort of structure, either for defence or thermoregulation (as hypothesized for prothoracic lobes)? Why would an adult palaeodictyopterid need tracheal gills? Surely their wings would have been a major hindrance to an aquatic lifestyle.

2) Page 10, line 225. Certainly it is feasible that the gills of *Diamphipnopsis* could allow it to breath while rowing on the water, thus is a survival strategy, however Palaeodictyoptera would not have been able to row. Palaeodictyoptera, like dragonflies, had two pairs of outstretched wings, which unlike stoneflies, they could not fold along their backs. A dragonfly, if it finds itself in the predicament of being on the water surface is likely to die. It can't swim and can't fly from the water surface because it's wings are trapped by the surface tension (see 'Drowning Dragonfly' on You Tube). It's either going to drown or be eaten by a fish. Its only hope is to drift to the water's edge and be able to grab onto something to drag itself out (I don't know if this has ever been observed or if a water-logged dragonfly would have the strength to do this). I can imagine that if an adult palaeodictyopterid found itself in a similar predicament it would suffer a similar fate. The only possible advantage I can see to it having lateral gills is that while floating on the water surface it may be able to get more oxygen to prevent it from drowning and thus give it more time to drift to the water's edge and to safety (if it had the strength to pull itself out and if it didn't get eaten by a fish or amphibian first- there were some big ones around at that time).

Author's Response to Decision Letter for (RSOS-190460.R1)

See Appendix B.

Decision letter (RSOS-190460.R2)

09-Aug-2019

Dear Dr Prokop,

I am pleased to inform you that your manuscript entitled "Ecomorphological diversification of the Late Paleozoic Palaeodictyoptera reveals different larval strategies and amphibious lifestyle in adults" is now accepted for publication in Royal Society Open Science.

on behalf of Kevin Padian (Subject Editor)
openscience@royalsociety.org

Appendix A

Response to Referees

Reviewer: 1

This is an interesting manuscript, sometimes the conclusions are very bold and some are far-fetched. I would suggest the authors read through it and may try to qualify their most important statements.

OK, we have checked and slightly improved our statements in the final ms.

Normally, I'm not a big fan of stating the English needs revision because I'm not a native speaker myself. However, I feel like there are some parts with better and some with worse English. It feels like the last revision was made by a non-native speaker and some of the changes stand out. Since there is a native speaker under the authors I would suggest him to give the manuscript a thorough read.

The final version of article has been once again checked by a native speaker.

However, generally I suggest publication after minor revision. Further comments are in the following:

Abstract

P7 L19 and following - I would hardly recommend avoiding terms like superorder or other ranks, we live in a time after the brilliant work of Henning so please act like it.

In general I agree with reviewer to use carefully the higher rank taxa. But I think in this case it makes sense to specify the rank of Palaeodictyoptera as number of potential readers will not be familiar with this spectacularly diverse extinct group of insects.

P7 L24 - "number of mature nymphs" is very unspecific

OK

P8 L42 - "stem groups" I would prefer stem group representatives. However, the sentence is monstrous can you may divide it?

We think this sentence is fine.

P8L47-48 - the transition from habitat to predation mode is rude ("... prior to the late Permian [6,7]. Wootton [6] considered ...") please add a nice connecting sentence.

OK

P9L75 - "... not demonstrably terrestrial, but lacking aquatic adaptations" if this is a direct quote you would need the page number I suppose.

OK, thank you.

P9L76 - "A weak position ..." your absolutely right, but that's your interpretation of data, please leave this for the discussion.

We think it is fine on this place.

P9L81 - "morphologies" morphological adaptations?

OK, Thank you.

P9L81 - " The resolution of ..." this is an enormous sentence and very hard to follow please explain and simplify.

We have checked the sentence once again.

P10L86 - " This matter is made all the more interesting ..." What?

OK, corrected

P10L98 - And? What are your questions? Any hypothesis? What can we expect in your paper?

OK

There is a substantial part of the introduction missing! Maybe you can find parts of it in the very first part of the results section ...

We rather prefer the current structure of sections.

Method section

P10L101 - " ... comparative morphological study ..." that's not a method, you compared the morphology using stereo microscopy

OK, Thank you.

P10L103 - " Observations were made using a Zeiss Discovery V20 and Nikon SMZ1500 in dry state or under a film layer of ethyl alcohol." That sounds like you are using your microscopes under a film layer or ethanol.

OK, corrected.

Results/Discussion

P12L140 - "rudders" well if you see Zygoptera swimming the main power for the protraction is generated by the caudal gills. Of course, they use them for steering but not at all exclusively. Furthermore, there are numerous studies showing that Zygoptera do quite well without their gills.

OK

P12L143 - "Recent" recent

OK

P12L147 - " The onisciform nymphs ... inhabiting fast-flowing streams." citation is needed.

OK

P12L150 - " However, a similar habitus ... (Lampyridae:Duliticola sp.)." citation is needed.

OK

P13L183 - " [38,39]: see figures 1a,b]" something is wrong with the brackets.

OK

P14L208 - " These spinose structures on the gills serve to anchor the nymph beneath rocks in stony upland streams and for protection. citation is needed.

OK

For the last part of the discussion the paper on Ephemeroptera larvae of Ditsche-Kuru et al. 2010, JEB (doi: 10.1242/jeb.037218) might be of interest.

OK, this is an interesting point. Thank you

Figure 1 - I'm missing a scale bar, please add one if possible.

These drawings/ reconstructions are not in scale because they are illustrative only. We think it would be strange to have scale bar for each.

Reviewer: 2

Comments to the Author(s)

I read the comments of the previous referees 1 and 2 and I think that the authors improved the manuscript attending them. I only want comment about one response of the authors to a topic noted by the both referees. Authors indicated that: "... a banded pattern of coloration is not indicative of only aquatic environment ... that contemporary early instar nymphs of Odonata bear exactly the same circular banded pattern on wing pads which serve for their protection against the predators. In later instars this pattern disappears because nymphs are larger and having fewer risk of attacks. This can be seen on early instar nymphs of Palaeodictyoptera, so we think it is important to demonstrate this evidence in our text." Similar explanation was provided to referee 2. However, please note that this is a strongly "adaptationist explanation", but it can be analyzed under the constructional morphology model (e.g., see Briggs, D.E.G., "Seilacher on the Science of Form and Function"), mainly considering that this character is present in Palaeodictyoptera and other related groups during early instars. Maybe the authors could include a sentence about additional potential explanations. For example, it is not plausible that this pattern was highly-energy-consuming to maintain it during ontogeny, and it is not plausible that the hypothetical decrease of predation in mature instars implicated a high energy savings that would result, from natural selection, not to maintain it.

Yes, it is interesting point. But we are not completely sure.

The main comment I want to include is that in this manuscript there is not mention of taphonomical evidence supporting or not the main topic of this manuscript. It is know that in several burial environments there is an overrepresentation of aquatic taxa, and etc. At this respect, I think that one of the authors could include some comments about this topic considering the fossil record of Palaeodictyoptera in general, because he investigated about taphonomy in aquatic environments (André Nel).

Yes, it is good point. We have included a paragraph concerning the taphonomy. However, it is rather difficult to find convincing support as these immature insects and their exuvia are scarcely found in contrast to adults. Also they can be also easily secondarily transported by wind and water currents.

Minor corrections:

Line 5: “Engel,” changes to “Engel”

OK

Line 45: Lagerstätte” (outcrop in German) is not informative. Do the authors want to indicate “Konzentrar-Lagerstätte” or “Konservat-Lagerstätte”? I think that the latter.

OK

Line 75: “....” Changes to “...”

OK

Lines 93-94: “...genus, Dunbaria Tillyard (Spilapteridae), uncovered...” [note the two commas]

OK

Line 129: maybe figures (plural). Please, revise this detail along the manuscript because it is not clear the guideline applied

OK

Lines 131-132: “...onisciform body form...” changes to “...onisciform body...”

OK

Line 134: “...Yorkshire, UK, assigned...” [note the extra comma]

OK

Line 144: “have” changes to “has”

OK

Line 170: please, revise “...this could have been be...”

OK

Line 204: “... a similar nine pairs...”, please revise the use of singular or plural in this expression

OK

Line 226: “reveal” changes to “reveals”

OK

Line 245: “follwing” changes to “following”

OK

Lines 256-260: drawings and reconstructions are always interpretative. I think that this section must to include this contribution. At this respect, is evident that MP contributed, but who is ZC (apparently not a co-author) as indicated at the end of figure caption 1? If is not a co-author, please include his name in Acknowledgements.

OK

Line 382: “Bull.” in italics

OK

Figure captions: the genus and species names are not in italics!

OK, it was our mistake in the previous version.

Figure caption 1: “U.S.A..” changes to “U.S.A.” (two times). Note that for the same, the authors used indistinctly U.S.A. and USA. In line 27, apparently authors must indicate if the specimen is mature (subfigure c)

OK

Figure caption 3: line 29: (d,e) maybe corresponds to (d,f), please revise it. It is not clear the taxon of subfigure f

OK

Figure caption 4: line 24 the indication of subfigure a must be in parenthesis. Line 25: M.P. is the same author that MP in figure caption 1. Please, use the same format. It is not indicated that (c,d) are photographs of the same specimen that in (b). I deduced it.

OK

Figure caption 5: in the title, please include “fossil and Recent” even that circumstance is evident. Line 33: the indication of subfigure a must be with parenthesis. The explanation of the subfigure (e) lacks indication of the outcrop and country.... Note that the correct spelling is “La Araucanía” (two times in this figure caption)

OK for all

Appendix B

Response to Referees

Reviewer: 3

Comments to the Author(s)

This is an exceptionally interesting topic. The findings and interpretations are of great relevance for the understanding of the early evolution of pterygote insects. The authors are obviously very competent.

I. 88: I would reword this to make it less ambiguous (“Palaeoptera problem”).

OK, thank you.

. 97: “uncovered a mosaic of characters and numerous homologies to Odonatoptera, Ephemeroptera, and also Neoptera [27].”

What exactly means this, the identification of homologous structures? Or potential synapomorphies? Please clarify.

OK, we clarified this.

I. 146: “...microstructures as setae...”

Something seems to be missing in the sentence.

OK, thank you.

I. 151. “damselfly nymphs”

These immatures have specific larval structures like the labial mask and the terminal appendages, so they are by definition larvae and not nymphs.

OK, we changed for damselfly larvae and also replaced nymphs for larvae throughout the text.

I. 157. Also in some larvae of Scirtidae and Torridincolinae (e.g. Beutel et al. 1998 or Handbook of Zoology Coleoptera, Vol. 1)

Yes, we have included the link to these groups. Thank you.

I. 199. This statement is awkward. It is evident that this stonefly species is not closely related to the extinct taxa under consideration. So it is obvious that the stonefly only indicates that this scenario is possible on principle.

Yes, we agree this group is not closely related to Palaeodictyoptera, nevertheless the phylogenetic analysis of Sroka et al. (2015) resulted in sister relationships between Palaeodictyoptera and Neoptera. We mainly aimed at the demonstration of an example from extant insect group bearing functional abdominal gills in adults. We like to point out their lifestyle and specializations without clear reference to their relationships.

As a whole this a well written study with very interesting results. In the present version the interpretations are presented carefully and cautiously. I recommend publication after very minor revision.

I recommend a final linguistic check by Dr. Engel.

OK, the manuscript has been checked once again by Prof. Engel.

Reviewer: 4

Comments to the Author(s)

This is an interesting and thought provoking article that deserves publication, however there is a problem that the authors and reviewers have not addressed. I'm perfectly happy that palaeodictyopteroid nymphs could have been aquatic however I'm not convinced that adults could also have been amphibious and survived in an aquatic/semiaquatic environment.

1) Page 9, line 207. Why are the lateral structures 'tracheal gills'? What's the evidence? Why could they not be another sort of structure, either for defence or thermoregulation (as hypothesized for prothoracic lobes)?

Thank you, these are relevant points for the discussion. First of all these lateral abdominal structures are not tergal as their basal parts are overlapped by terga and they are probably emerging between the two corresponding segments. Second, they are distinctly bifid-like as tracheal gills in many and also nymphs of mayfly stem *Protoreismatida*. Of course, we cannot definitely exclude their different function as for defense or thermoregulation, but we would rather expect those defensive or thermoregulatory structures should be fixed and rigidly connected with the tergum or sternum. But we provided note for this alternative interpretation.

Why would an adult palaeodictyopterid need tracheal gills? Surely their wings would have been a major hindrance to an aquatic lifestyle.

It is good question, but we suspect the adults were probably living close with their larvae in aquatic or semiaquatic environments, perhaps much like some recent stoneflies (e.g., *Diamphiphnopsis* or *Pteronarcys*). We can envision that some of these insect species were living in a film layer of water edges or in waterfalls for instance. Yes, the wings of megasecopteran *Corydaloides* were outstretched at rest, but strongly narrow near the wing bases in comparison to dragonflies (Anisoptera).

2) Page 10, line 225. Certainly it is feasible that the gills of *Diamphiphnopsis* could allow it to breath while rowing on the water, thus is a survival strategy, however Palaeodictyoptera would not have been able to row. Palaeodictyoptera, like dragonflies, had two pairs of outstretched wings, which unlike stoneflies, they could not fold along their backs. A dragonfly, if it finds itself in the predicament of being on the water surface is likely to die. It can't swim and can't fly from the water surface because it's wings are trapped by the surface tension (see 'Drowning Dragonfly' on You Tube). It's either going to drown or be eaten by a fish. Its only hope is to drift to the water's edge and be able to grab onto something to drag itself out (I don't know if this has ever been observed or if a water-logged dragonfly would have the strength to do this). I can imagine that if an adult palaeodictyopterid found itself in a similar predicament it would suffer a similar fate. The only possible advantage I can see to it having lateral gills is that while floating on the water surface it may be able to get more oxygen to prevent it from drowning and thus give it more time to drift to the water's edge and to safety (if it had the strength to pull itself out and if it didn't get eaten by a fish or amphibian first- there were some big ones around at that time).

OK, in general we agree with this comment and getting more oxygen when floating on the water surface is another possible function. The video of water-logged dragonflies is very convincing, but the wings in *Corydaloides* are slightly different in form from these dragonflies. Furthermore, we think that in other stoneflies like

Neuroperla schedingi or *Diamphipnoa annulata* their gills on adults are not necessarily function for rowing. Hence, if these structures in *Corydaloides* are functional gills we rather suspect the lifestyle in water edges or film layer - waterfalls instead of deep pools. Another crucial point is whether these supposed gills were functional or just rudimentary ... These structures persisting in adults of many insect species with aquatic larvae as already documented by Štys and Soldán (1980), but their function is not well explored.